# Finite Element Modelling and Experimental Validation of Scratches on Textured Polymer Surfaces

**DOI:** 10.3390/polym13071022

**Published:** 2021-03-25

**Authors:** Weimin Gao, Lijing Wang, Jolanta K. Coffey, Hongren Wu, Fugen Daver

**Affiliations:** 1School of Engineering, RMIT University, Melbourne, VIC 3001, Australia; henry.wu@rmit.edu.au; 2School of Metallurgy and Energy, North China University of Science and Technology, Tangshan 063600, China; 3School of Fashion and Textiles, RMIT University, 25 Dawson St, Brunswick, VIC 3056, Australia; Lijing.wang@rmit.edu.au; 4Ford Motor Company, Research and Engineering Product Development Centre, 20901 Oakwood Blvd, Dearborn, MI 48124-4077, USA; jcoffey4@ford.com

**Keywords:** textured surface, scratch, scratch resistance, thermoplastic polymer, finite element modelling

## Abstract

Surface texturing is a common modification method for altering the surface properties of a material. Predicting the response of a textured surface to scratching is significant in surface texturing and material design. In this study, scratches on a thermoplastic material with textured surface are simulated and experimentally tested. The effect of texture on scratch resistance, surface visual appearance, surface deformation and material damage are investigated. Bruise spot scratches on textured surfaces are found at low scratch forces (<3 N) and their size at different scratch forces is approximately the same. There is a critical point between the bruise spot damage and the texture pattern damage caused by continuous scratching. Scratch resistance coefficients and an indentation depth-force pattern are revealed for two textured surfaces. A texture named “Texture CB” exhibits high effectiveness in enhancing scratch visibility resistance and can increase the scratch resistance by more than 40% at low scratch forces. The simulation method and the analysis of the power spectral density of the textured surface enable an accurate prediction of scratches.

## 1. Introduction

Surface texturing is a common functionalization method for altering the surface properties of a material or component. The most successful application of surface texturing in engineering is in the improvement of tribological performance [1,2,3,4,5]. Its applications have rapidly expanded, along with contributions in various fabrication techniques [6,7,8,9,10]. This paper focuses on producing hard textured polymeric parts with good scratch/mar resistance and durable gloss levels through surface topography modification.

The texturing of polymeric parts can be achieved either via direct patterning or by replicating the surface of a tool with a specified topography, such as wood or leather [11,12]. It was found that certain textures could increase scratch resistance and conceal the surface damage of polymeric materials [13,14]. This has led to an increased focus on understanding how surface texture influences surface damage features, and scratch resistance and visibility [13,15,16,17,18]. The surface damage features demonstrate both brittle and ductile modes of deformation which are sensitive to the contact properties in scratching [19,20]. The contact features such as the force magnitude, the normal versors, and the principal curvatures greatly depend on the contact point, when the size of the scratch tip is lower than the grain size of the rough surface or textured surface. The scratch resistance and visibility on polymer surfaces are other major concern areas of material scientists. These are complex issues that involve the physical deformation of material, the surface damage features and the surface optical properties. These features become more complicated when scratching on rough and textured surfaces is considered. There are an extremely large number of surface textures being used in various materials and products for different purposes, such as improving tribological performance, scratch resistance, thermal resistance, visual appearance, additive manufacturability, wettability, biocompatibility, and light absorbability.

The load scratch test is an effective and widely used method to assess surface texture and investigate the failure of substrate material. Although laboratory-based experimental testing can be used to evaluate the capability of an existing texture in terms of scratch resistance and material damage [15,21,22], it is a time-consuming and capital-intensive process for a new texture; for instance, examining a new texture on a thermoplastic polymer plaque includes mold design and making, material preparation, sample injection and cooling, scratch test, and analysis of results. Most existing surface damage models consider the scratch force, material properties, and the viscoelastic–viscoplastic behavior of material in the case of polymeric material scratches [23], but seldomly take into account the roughness and texture of the surfaces. The prediction of a new surface texture on the scratch properties of the material is also of significant interest to both academia and the manufacturing industry for scratch management. Computer simulation is a straightforward method for predicting the scratch properties of materials. However, a literature review conducted by the authors shows that only a few simulation studies are focused on the influence of surface texture. Almost all simulations are only for smooth surfaces without large material deformation and damage, and are aimed at studying the material behaviors during indentation [24,25] and scratching [26,27,28,29,30,31], the influence of material mechanical properties (elastic modulus, yield strength, Poisson’s ratio, coefficient of friction) [17,32,33,34,35,36,37], type of materials [24,26,27,38], surface texture [22,39,40,41] and roughness [42,43], interface interaction between rough material surface and scratch tip [44,45,46,47,48], and scratch visibility [18,49,50].

Textured surfaces render scratch simulation more challenging, adding to the difficulty of simulating large deformations and damage of materials [39,51]. The failure mechanisms of materials, e.g., cracking, chipping, and delamination, will greatly depend on the local contact behavior. They may include continuous deformation of the roughness zone and substrate and the deformation of only asperities. The apparent friction coefficient also varies during scratching when the characteristic wavelength of the texture is larger than the size of the scratch tip, due to the change of the ratio between the tangential force and the normal load.

In order to determine the scratch simulation of a textured surface for the investigation of a certain problem, the four characteristic lengths are associated, namely the scratch tip size, the wavelength of the texture pattern, the roughness (micro and nano roughness) of the surface, and the wavelength of the waves involved in the problem or the surface function. In the case of studying scratch visibility, the roughness should be considered in the simulation as it governs the surface–object contact and flow of light over the scratched surface. Both influence the visual appearance of the scratched surface. Contact interaction occurs between rough surfaces. In the case of most interfacial phenomena, the accuracy of simulation relies to a large extent on the accurate description of geometrical contact between surfaces of any given roughness [52]. In the case of polymer scratch, it is associated with appropriately defined friction properties. In finite element method (FEM) simulations, the influence of surface roughness on light flow that manipulates the visual appearance of the rough surface has not yet been discussed. It is desirable to compute the photorealistic appearance of the surface in the real world and, thereafter, study the ability of the texture to conceal scratches and the behavior of textured surfaces in reducing the visibility of scratches. The human ability to identify scratch damage relies on the visual capacity of the eyes. A human eye can identify a spatial pattern separated by a visual angle of 1/60° [53], at which angle the viewing distance must be less than 10 cm for an object of 30 μm or a resolution of ~900 dpi to be discerned. For instance, the size of roughness of an interior trim polymeric component used in automobiles can be as low as 20 μm. The change in surface topography that occurs at low scratch impacts is at such a level of asperity. It is well-known that human eyes show different color sensitivities [54]. The human visual perception is also highly sensitive to the structural information in a scene; it can extract information from similar textures and similar structures [55,56]. Therefore, studies on the scratch visibility of polymers [15,18,35] show that different textures possess different visibility resistance.

The features of scratches on the surfaces with different texture patterns have been systematically studied experimentally [13,15]. It was found that texture pattern significantly influenced scratch resistance and visibility. Our previous study of scratch modelling on smooth surfaces [30,39] demonstrated the large dependence of scratching process, material deformation and material damage features on contact properties, indicating that the scratch process and material failure mechanism on a textured surface would be very different to that of scratching on a smooth surface. To study the surface deformation and damage, the failure mechanisms of materials, and the functions of textured surfaces subjected to scratching, scratch simulation needs to be carefully constructed to capture all essential aspects related to the problem. In this study, we present simulation and experimental results of scratches on thermoplastic plaques with different textured surfaces and discuss the effect of texture, as well as visual scratch identification. A simulation method and the knowledge of the power spectral density (PSD) of the textured surfaces were used to accurately predict scratches and to examine the performance of the model in capturing their texture patterns and roughness features. The simulation results are evaluated via scratch experiments at different scratch forces and show accurate prediction of scratches for commercial surface textures.

## 2. Experimental

The samples used are injection-molded thermoplastic plaques. The material is a commercial polypropylene product (ADX5017 supplied by Advanced Composites, Inc., Ohio, USA). The plaques were molded with different surface textures.

The power spectral density (PSD) was used to analyze the surface roughness of the plaques. PSD is a fractal method that is considered as one of the better characterization techniques of surface topography and has been used by different researchers [57,58]. A detailed discussion of PSD can be found in the references [59,60]. In brief, a surface PSD is the Fourier transform of the autocorrelation function of the 2D surface height data, decomposing the rough surface into contributions from different spatial frequencies (wavevectors). The radial average on the Fourier transform of surface topography is generally used for isotropic surfaces and a 2D power spectrum, 2D-PSD, is obtained.

The PSD provides a representation of the amplitude (C) of surface roughness as a function of the spatial frequency (*q*) of the roughness. The PSD can also reveal possible periodic surface features of the texture and how the features are distributed. Figure 1 shows how the PSD of a rough surface changes when it includes a texture. The left and right PSD peaks, at *q* = 3 × 10^3^ − 1 × 10^4^ m^−1^, of the texture and the synthesized surface in Figure 1d are the representation of the texture distribution in X and Y directions, respectively. In general, the area under the PSD curve is equal to the square of the root mean square roughness ($R_{RMS}^{2}$, with unit of m^2^), i.e., the square of the standard deviation of surface heights (σ^2^) for the present case. Higher PSD values associate with rougher surface geometries. The spatial frequency of the roughness (wavevector: Q) is the inverse of the wavelength of the roughness features. The slope of the decay at higher spatial frequencies is determined by the Hurst exponent (H), which is directly related to the fractal dimension of a surface topography. A Hurst exponent with higher values indicates a smoother trend, less volatility, and less roughness [61].

Scratch tests on the plaques were conducted with the TABER^®^ multi-finger scratch/mar tester (Model 710). The textured surfaces and scratched surfaces were captured using a Cannon SLR camera equipped with a macro lens. The camera was tilted at a fixed angle from the surface normal, and a fluorescent light source placed opposite to the camera was adjusted to best highlight the scratches.

To evaluate the simulation results, the laboratorial scratch tests were conducted on different textured surfaces at fixed loads of 2 N, 3 N, 7 N, 10 N and 15 N using a steel scratch finger with a spherical tungsten carbide tip, 1 mm in diameter, and scratch speed of 100 mm/s. Images of the scratched surfaces were captured using a high-resolution camera under a light source that was adjusted to render the scratches on the surfaces noticeable. An image processing and analysis (IPA) program was developed with MATLAB and the Image Processing Toolbox to automatically identify the scratches from the textured patterns and parameterize the features of each scratch. The IPA system consists of six modules: image preparation, integration, background filtration, pattern filtration, scratch identification, and scratch characterization. The image preparation is designed to select a region for analysis and adjust the selected section if it is distorted. The image intensity is then integrated along defined coordinates. The background and pattern are filtered to eliminate the deviation of light source influence over the panel and the intensity response to the panel patterns. The identification and characterization of scratches are based on the peaks of the intensities of the filtered image data.

For instance, Figure 2a shows the image of the SB texture and five scratches on it, with the positions of the detected scratches highlighted using the IPA program. The background of the image was filtered. The intensity of each pixel in contrast to the background was then measured to calculate the features of the scratches, as shown in Figure 2b. The larger variances in pixel intensity clearly show the positions and sizes of the five scratches. The values of the differences in intensity were used to evaluate the visibility of the scratches.

## 3. Textured Surface Model and Scratch Modelling

### 3.1. Textured Surfaces

Figure 3 shows two different textures. The height profile of a small surface section of each texture was also presented to identify the features of the surfaces. The textured surfaces consist of primary patterned coarse grains and random small peaks and pits on the grains. The shallow bump (SB) texture (Figure 3a) is comprised of worm-like islands with fine peaks. The coarse bump (CB) texture (Figure 3d) consists of large and tall islands with small craters densely distributed over the island surface and small bumps in the basin. The features of them are summarized in Table 1.

The radially averaged 2D PSDs of the textured surfaces are given in Figure 3c,f, revealing the inherent characteristics of the coarse grains and surface roughness. The curves of dx = 18 μm in Figure 3c,f show the PSDs of the original SB and CB textures, respectively. The data at about *q* = 2 × 10^3^ − 2 × 10^4^ m^−1^ in Figure 3c represent the size and distribution of the islands in the SB texture. The wide range of *q* indicates that the islands are long with varying lengths. The islands are worm-like and have average thickness of 213 μm. The spacing between islands is about 460 μm. At *q* = ~2 × 10^4^ m^−1^ the Hurst exponent H changes. H is low at *q* > 2 × 10^4^ m^−1^, while H becomes high at *q* = 2 × 10^4^ m^−1^. The drop of the PSD at low *q* (<10^4^ m^−1^) approximately stops at *q* = ~2 × 10^3^. These are attributed to the contribution of the worm-like islands in the SB texture. The obviously high values of C^2D^ at about *q* = 2 × 10^3^ m^−1^ in Figure 3f indicate the features of the large islands of the CB textures. These islands are approximately 120 μm high and 700 μm in width. The obviously high PSD profile at *q* = 1.3 × 10^3^ − 5 × 10^3^ m^−1^ is attributed to the large and tall islands. The contribution of the islands is much similar to the influence of a texture on a rough surface illustrated in Figure 1. The PSD at higher *q* reveals the features of the small peaks, craters, and bumps. A close comparison of the PSD data for *q* > 2 × 10^4^ m^−1^ in Figure 3 reveals that they are approximately the same, indicating that the peaks in the SB texture and the craters and bumps in the CB texture have approximately the same characteristic size and distribution features over the surfaces.

### 3.2. Surface Models

The meshing of materials is a critical step in simulations employing the FEM. Unlike the meshing of materials with ideal smooth surfaces, whose geometrical features would be fully obtained if an appropriate mesh size were to be adapted based on the grid independent of the analysis of the simulation results, the meshing of rough surfaces becomes an important issue when the interface interaction is significant, such as, in addition to the modelling of scratch and material damage that will influence the visual perception of the polymer surface in the present work, the modelling of wear, friction, contact electrical conduction, contact thermal conduction, and fluid transport over rough solid surfaces [45,52,62]. It is difficult to build a model of a real rough surface and, generally, the rough surface is simplified to save simulation costs, including the requirement for hardware and computing time. One simplification is to describe the rough surface as consisting of larger hills and valleys, called asperities, neglecting the smaller rough peaks and pits. In other words, the contact interaction is that of asperity against asperity. In the scratching of the textured polymeric surfaces shown in Figure 3 with a spherical tip, the interaction is that of contact between a smooth sphere and a rough surface. When the scratch tip size (for instance, diameter of 1 mm) is much greater than those of the small peaks and craters (for instance, the root mean square height of the SB and CB rough surfaces, *h*_RMS_ < 20 μm, when eliminating the contribution of the islands), the textured surfaces, as illustrated in Figure 3, can be filtered to only include the primary patterned islands in the simulation of scratches on polymeric plaques if the aim is to study material deformation (scratch formation) and failure, or to examine the material properties, texture pattern, or scratch load [39]. However, the optical properties of the scratches cannot be accurately studied, as the smaller peaks, craters, and bumps play an important role in influencing the flow of light.

A rough surface model that accurately represents the real textured surface is essential for the scratch simulation to yield results closer to the actual changes in surface topography. Therefore, the element size for meshing the surface must be carefully determined. The model should capture all the surface features that affect the transport of light. Meshing very small peaks and tips would result in exponential increases in the number of elements and computing time. Employing unstructured surfaces and volume meshes would help capture more features of rough surfaces.

In this work, a three-dimensional laser scanning system was used to capture the surface geometry of the plaques and the standard triangulation language data was converted to a 2D surface height matrix, Z(x, y), via MATLAB. The rough surface models were then generated based on the measured surface height matrix, Z(x, y), of the SB and CB textures. The textured surfaces were tested using different mesh sizes and were analyzed via the PSD method, as shown in Figure 3. The results showed that when the element size was smaller than the resolution of the scanned surface (e ≤ dx), most features of the textured surfaces could be captured. At e = 20 μm and e = 30 μm, the Hurst exponents, Hs, are the same, but *q*_S_ reduces with the element size. This indicates that, when a 20 μm mesh was used, only very small surface roughness features were filtered out. When e > dx, an increasing number of small-scale roughness features was lost with an increase in mesh size. When a 30 μm mesh was used, the tops of the peaks on the worm-like islands in the SB texture and the roughness of the small craters over the large island surface and smaller bumps in the basin of the CB texture were filtered out. Hurst exponent H increases when the element size is larger than 30 μm, i.e., H (e = 100) > H (e = 50) > H (e = 10–30). This indicates that the surfaces become smooth with the increase in mesh size. When larger meshes (50 to 100 μm) were used, the small peaks, craters, and bumps in both the textures were completely removed. A few features of the worm-like islands in the SB texture were additionally filtered, as shown in Figure 3c, when the mesh size was increased to 100 μm. In short, knowledge of the PSDs of the samples and meshed surfaces can aid in the quantitative examination of the mesh size required to capture the desired features of the rough surface. For the textured surfaces used in this work, meshes smaller than 30 μm can capture the small roughness features. Meshes larger than 50 μm are not recommended for meshing the surfaces, as this would entail the loss of texture patterns to a certain extent, especially in the case of the worm-like islands in the SB texture.

### 3.3. Scratch Modelling

In the scratch simulations, the material is described as a hyperelastic and viscoelastic model, which was generated from its compression strain-stress curve by MCalibration. The model is a parallel rheological framework (PRF) model, which uses a Yeoh hyperelastic model for the elastic behavior of the polymer and three power-law strain flow models for modeling the viscoelastic behaviors [63]. The four networks connect in parallel. The description of the PRF model and its formulation are given [64] and the parameters are listed in Table 2.

The model was also compared to an elastic-plastic with isotropic hardening (EPI), where the plastic model was described by 10 terms to best match the experimental data set [30]. The square of the correlation between predicted stresses and experimental stresses, R^2^, was calculated as the evaluation parameter and it shows R^2^ = 0.984 for the PRF model and R^2^ = 0.997 for the EPI model, respectively. The different material models have been compared for their application in modelling polyethylene terephthalate, high-density polyethylene, polytetrafluoroethylene and other polymers [65]. It shows that the parallel network model with two power flow networks and the elastic-plastic with isotropic hardening model have approximately the same model calibration error when they are applied for thermoplastic polyester.

The scratch simulation consists of two steps, as shown in Figure 4—the indentation process (step 1) and scratching (step 2). The scratch tip in contact with the material surface was first pushed down. The vertical force was linearly increased in intervals of 0.05 s up to a specified force (from A to B). The tip then moved horizontally over the rough surface along the Y direction at a speed of 100 mm s^−1^ (travelling 6 mm in 0.06 s) at the specified force (from B to C). Then, the load-controlled scratching covered more than 10 large grains of the SB texture and more than two large islands of the CB texture. The scratch started at the point of X = 2.5 mm and Y = 1 mm. The scratch tip in the X direction was restricted such that straight scratches were produced. The bottom faces were fixed, and zero displacement was applied for both +Y and −Y boundaries in the simulation, while the symmetrical boundary condition was applied for the +X and −X faces. The simulations were performed for scratch forces that were the same as those applied in the experiments. In the simulation, the friction coefficient, *μ*, decays exponentially from the static value, *μ_s_*, to the kinetic value, *μ_k_*, according to the Formula:(1)$$\mu =\mu_{k}+\left(\mu_{s}-\mu_{k}\right)e^{-d_{c}s}$$
where *d_c_* is the decay coefficient, and *s* is the sliding velocity. The static friction coefficient, the kinetic friction coefficient, and the decay coefficient (*μ_s_*, *μ_k_*, and *d_c_*) were experimentally measured on a smooth surface plate of the same material as the rough surfaces. They are 0.102, 0.086 and 0.25, respectively.

The coupled Eulerian–Lagrangian (CEL) method [66,67] was employed for the scratch simulation considering the very complex surface textures and roughness, and the extreme material deformation and damage involved in the scratching, as shown in Figure 4b. This numerical technology combines the Lagrangian and Eulerian meshes in a single analysis, wherein the material inside the Eulerian region is quantitatively represented by the Eulerian volume fraction (EVF). The Lagrangian object can move inside the Eulerian domain and interacts with the Eulerian material when they come in contact. The interaction between Eulerian and Lagrangian meshes is described with contact algorithms [63]. For the fundamental of the CEL method, the reference [68] is proposed. For scratching on the polymer surface, the Eulerian material is the polymer. The plaque domain (the grey region in Figure 4) is discretized using the Eulerian mesh and also set as Lagrangian meshes, while the domain above the rough surface (the white region in Figure 4) is the Eulerian region only. The interface between these two kinds of meshes, i.e., between the scratch tip and the polymer surface, will change with the motion of the Lagrangian scratch tip. In the present work, the contact between the scratch tip and polymeric material was expressed using a general contact algorithm based on the penalty contact method. The CEL approach combines the advantages of Lagrangian analysis in deformable mesh and Eulerian formulation in the spatial movement of continuum, and this has been demonstrated via the simulation of materials undergoing large deformations [69,70,71] and its merits and shortcomings in the application to model the deformation of polymers, compared to the Abaqus/Explicit arbitrary Lagrangian–Eulerian (ALE) adaptive meshing method, were discussed by Gao et al. [30].

## 4. Results and Discussion

### 4.1. Features of Modelled Texture Surfaces

A model with a surface area of 5 mm × 8 mm and thickness of 3 mm was created for each plaque, as shown in Figure 5. The scratch was performed on a 4 mm × 7 mm textured surface. The plain area extending from the rough surface makes the plaque model sufficiently large such that changes in strain/stress during scratching occur only within the center and do not spread to the boundaries of the model. Contrarily, meshing can be easily handled to reduce the number of meshes.

The modelling results of e = 30 μm in Figure 5 show that the model for the SB texture captured the worm-like islands and their peaks. As shown in Figure 5a (original and model), the model includes the islands and all the obvious peaks on each island, but some sharp peaks and dips were filtered out as indicated by the Z-scale, which reduced from the original scale of +0.06 mm to −0.06 mm to the scale of +0.04 mm to −0.04 mm. As shown in Figure 5b, only the sharp dips of the CB textured surface were filtered. The Z-scale changed from the original scale of +0.06 mm to −0.08 mm to the scale of +0.06 mm to −0.06 mm. Figure 3 shows that the model for the CB texture captured the islands, craters on the top of the islands, and peaks in the basin. Comparing the modelled textures with the texture surface images as presented in Figure 3, it can be concluded that the scale of roughness replicated in the models is identical to that captured by the high-resolution digital camera. This is important in studying the effect of the textured surfaces on the visibility of scratches and damage. The characteristic size of the surface grains captured is much lower than the identifiable size by a normal human eye at a viewing distance of less than 10 cm. The change in roughness that can be resolved by the human eye at very small viewing distances can, therefore, be simulated using the model.

Figure 5 clearly shows the aforementioned texture pattern features. The worm-like islands in the SB texture and the small craters on the top of the islands and small bumps in the basin in the CB texture are more clearly seen in Figure 5 than in Figure 3. The worm-like islands have a wide range of lengths, which are represented by the PSDs at *q* = ~2 × 10^3^ − ~2 × 10^4^ m^−1^ in Figure 3c, as described above. Their width and spacing are small and approximately the same. These features are represented by higher PSDs at a spatial frequency of ~2 × 10^4^ m^−1^.

### 4.2. Indentation and Scratch Resistance

Figure 6 shows the displacement of the indenter at different forces and at different force-loading rates. The data were from different indentation processes, i.e., the step one in the scratching simulations, where the force was linearly increased. The surface here refers to the median plane of the surface texture, so that the initial point of contact (IPC) locates at the top of the surface, showing a negative distance from the sample surface in Figure 6. Note that IPC depends on the position of the indentation in relation to the texture pattern. The indenter moved through the textured surface of 0.08 mm in thickness and reached the substrate surface at a force of about 4 N. The displacement is shown as a nonlinear function of the force in this step. The surface deformation or damage caused by the indenter would not lead to a visually observable change of the surface appearance. The typical linear indention regime was finally reached with a further increase in the load, as indicated in the figure.

The scratch resistance was calculated based on the forces in the scratching step of the simulations. The resistance coefficient (µ_R_) is defined as the ratio of the force in the scratching direction to the scratch force. Figure 7a shows the instantaneous µ_R_ of patten CB during scratching under different forces. The fluctuation of µ_R_ relates to the pattern and the kinetic energy of the indenter. Skipping of the indenter over the rough surfaces was observed. The zero values of µ_R_ at 2 N and 3 N are due to there being no interaction of the indenter with the surface. The indenter recontacted the surface at a high energy and a higher contact angle, leading to a normal contact force that is much higher than the specified scratching force. This resulted in the high values of µ_R_ at the recontact periods. It seems that there is a contact frequency that increases with the load. When the force approached 7 N, the contact became continuous and the main component of µ_R_ fluctuation was caused by the texture profile.

The resistance coefficients were averaged. Figure 7b shows the mean of µ_R_ of rough surfaces CB and SB to different scratch forces. The results show that the surface with texture CB has a higher scratch resistance than the surface SB. This has been demonstrated by the experiments and will be discussed in the following sections. To evaluate the function of the textures in increasing the scratch resistance, the results were compared to a smooth surface. The smooth surface shows a linear relationship of the resistance and the scratch force. It is interesting to note that texture SB resulted in a lower resistance to high scratch forces compared to the smooth force, but a higher resistance to low scratch forces. The most significant for the effect of the pattern is the substantially high resistance coefficient of surface CB, especially at low scratch forces. It is effective in increasing the scratch resistance.

### 4.3. Surface Appearance

The scratches simulated on the SB and CB textures and the experimental results are presented in Figure 8 and Figure 9, respectively. The simulated and test scratches appear the same. It is difficult for the human eye to discern the scratches at 2 N on the SB texture and at 2 N and 3 N on the CB texture from the rough textures. Moreover, the scratch at 2 N on the CB texture could not be identified, as any variation in the surface was obscured by the texture pattern and random roughness. However, it can be detected by analyzing the digital color information using the IPA technology [15]. Close observation, i.e., enlargement, of the scratches clearly shows some bruise spots among the textures. Although too minor to be observed by the human eye, they changed the profiles of grains and surface roughness, thereby influencing the flow of light and, as a result, rendering themselves detectable. At higher scratch forces (such as >10 N), continuous scratches were formed and significantly influenced the appearance of the structured polymeric surfaces.

It is well-known that the visual appearances of the textured surfaces and scratches depend on the lighting condition, colors of the plaque surface, position of observation, and performance and position of the camera. The simulations produced more identifiable and clearer scratches than the experimental tests, not only because the simulation provided the geometric profile of the scratched surfaces but also because the images of the simulated scratches were produced by computer rendering with the clearest lighting. This demonstrated the merits of the finite element method in modeling scratches and the importance of the simulation result rendering in the prediction of the visual appearances of scratches, especially if the simulated scratches are rendered under lighting conditions and image capturing parameters equivalent to experiments.

### 4.4. Surface Damage and Scratch Size

The features of the scratches on the textured surfaces were quantitively described by two parameters: Average width, $\bar{W}$ and transverse (the X direction in the simulation model) width, ${\bar{W}}_{T}$. The average width was calculated by dividing the deformed surface area by the length of the path of the scratch tip (L_S_), as shown in Figure 10a. The transverse width is the average of the maximum transverse dimensions (*W_T_*) of the scratched areas. The surface damage was described by the scratched volume, *Vs*, and the debris volume, *V_B_*, per length along the scratch, as shown in Figure 10b. The deformed surfaces, scratch volume and debris volume were determined by comparing the original and scratched surface profiles. The results are shown in Figure 11.

The simulation predicted the scratch profile and deformation of the material very well. At low forces, bruise spots on the surface grains were produced by the scratch tip, as illustrated by the scratches at 2 N, 3 N and 7 N in Figure 8, Figure 9 and Figure 12. The bruise spots are owed to the jump of the tip over the surfaces [72] and do not change the texture pattern, as shown by the enlarged scratches in Figure 8e and Figure 12. The scratched volume, V_S_, linearly increased with the force and no debris was produced at the low forces as shown in Figure 11b, (2 N–7 N). The micro roughness was smoothed over the bruise spots. The spots distribute linearly at a certain distance interval. An interesting finding is that the size of the spots, ${\bar{W}}_{T}$ in Figure 11a, at 2 N and 3 N are approximately the same, and larger than that at 7 N. The interval greatly decreased when the scratch force increased to 3 N from 2 N. Figure 7 gives the details of the interval. The average intervals at 2 N and 3 N are 1.348 mm (std = 0.159 mm) and 0.954 mm (std = 0.157 mm), respectively. Figure 11a shows that the SB texture resulted in a larger scratch width than the CB texture, as its grains are small, sharp, and more directional and orientational than the large CB grains. The scratch became nearly continuous at 7 N. The no-scratch area can also be found by closely examining the simulated scratches in Figure 12, but Figure 7 shows that the stage with zero resistance coefficient disappeared.

At high forces, the primary large grains and the substrate were damaged. The body of the worm-like islands in the SB texture and the top surfaces of the coarse islands in the CB texture were obviously damaged by the scratch tip at 10 N, whereas the material deformation degree at the bottom of the CB texture is very low. The experiments show that the damage of the substrate of plaque with CB texture is still slight at 10 N and 15 N, but large material failure occurred in the substrate of SB texture at the forces, due to the difference in the features of their grains. The grains of the CB texture are larger and rounder and have a less directional structure and less orientational distribution than the SB texture grains, as mentioned before. The simulation results show that the material damage extended to the valleys of the CB texture at 10 N. Figure 11 shows a fast increase in both the scratch volume and the debris volume with the force at the stage. The width of scratches on both textures rapidly increased as the scratching force increased from 7 N to 10 N. For the continuous scratches at high forces, their $\bar{W}$ is larger than their ${\bar{W}}_{T}$ due to the material deformation close to either side of the scratches and the transverse swing of them.

There is a critical force between the bruise spot damage and the texture pattern damage as shown in Figure 8, Figure 9, Figure 11 and Figure 12. The ${\bar{W}}_{T}$ of the bruise spots did not change with force, whereas $\bar{W}$ slightly increased. The ${\bar{W}}_{T}$ is the smallest when the scratch tip starts to creep on the rough surface. The energy is mainly consumed in friction and removing the micro roughness when the scratch tip creeps on the textured surfaces. Therefore, ${\bar{W}}_{T}$ is a useful parameter to identify the critical point between the scratching types. The pattern and substrate damages became serious with the increase in scratch force. The energy was mainly spent by the plastic deformation of the texture and the substrate.

The simulation predicted the scratch profile, the surface damage and the quantitative scales well at the two typical scratching regions. It appears that the prediction of scratch on the SB texture is more accurate than that on the CB texture. Figure 11a shows that the simulation predicted the sizes of the scratches well on both the textures. The widths of the scratches tested were calculated using the IPA method, which was based on the intensity of each pixel in the scratches in contrast to the background. The difference in width between the experimental and simulated scratches on the CB texture is small except at 7 N.

The simulated scratch on the CB texture is more obvious than the experimental scratch both in terms of material deformation and appearance. Figure 11a shows that the simulated widths are higher than the measured values, especially on the CB texture at 3 N and 7 N. The deviation is attributed to the differences between the simulation and experiment in terms of both scratching position and scratch finger behavior. The scratch tip in the simulation was located at the position where the tip moved over the tops of a few islands, whereas the scratch tip in the experiment mainly moved along the sides of a few islands and on the bottom/basin of the texture. The former resulted in larger deformation of the material, considering that the scratch tip, a perfectly rigid object, was restricted in the X-direction motion in the simulation (for the coordinate, see Figure 5). The scratch finger in the experiment could not, however, behave as such a rigid body because it had a certain length and neither the scratch finger nor the test plaque could be mechanically fitted on the experimental setup ‘perfectly’. The scratch tests were actually conducted using an elastic scratch finger with a rigid tip. The positions of the scratch tip and plaque varied in the X direction during scratching owing to the change in contact force with the position of interaction between the tip surface and grain surface. The slight swings of both the scratch finger and plaque led to crooked, i.e., not straight, scratches on the CB texture and a lower impact on the material than that caused by a perfectly rigid scratch tip.

## 5. Conclusions

The combination of simulations and experiments of scratching different textured surfaces with a spherical scratch tip provided a fundamental understanding of scratch behavior on textured polymeric surfaces. Jumping of the scratch tip over the textured surfaces and generation of bruise spots on the surfaces were found at low scratch forces. The continuous contact of the scratch tip with the textured surfaces occurred at only high scratch forces. The transverse widths of bruise spots at different scratch forces are approximately the same, but the interval between the spots decreases with the force. The transverse width is a useful parameter to identify the critical point between the bruise spot damage and the texture pattern damage caused by the continuous scratching. The critical force is about 7 N for the polypropylene.

The simulations quantitatively revealed the resistance coefficients of two textured surfaces and an indentation depth-force pattern. Texture CB showed high effectiveness in enhancing the scratch visibility resistance and increased the scratch resistance by more than 40% at low scratch forces (<3 N) and 9% at high forces (>7 N). The scratch visibility was strongly influenced by the surface texture. The scratches on the CB texture showed a low visibility, but a large size.

The simulation was based on the accurate capture of the textures and surface roughness features. Meshes smaller than 30 μm could capture the small roughness features. The experimental evaluation showed that the simulation predicted the appearances and profiles of the scratched surfaces well. This is significant in computing the appearances of scratched surfaces and in modelling the influence of textured surfaces on the scratch visibility resistance.

## Figures and Tables

**Figure 1 polymers-13-01022-f001:**
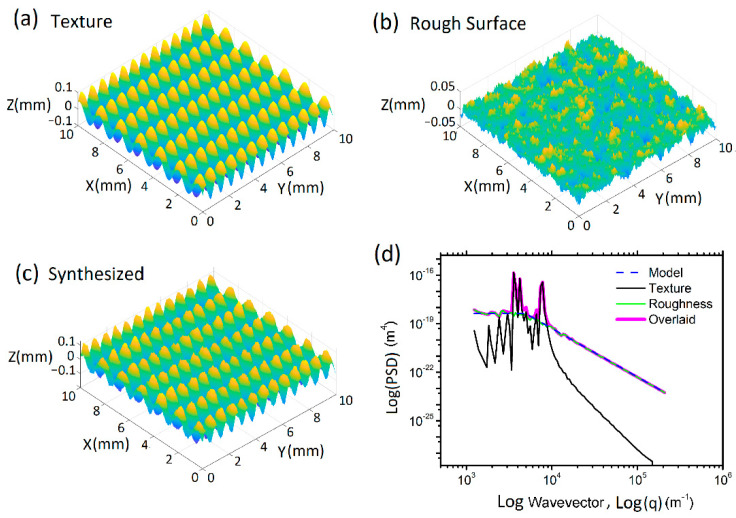
Example of PSD (**d**) for a textured surface (**c**) synthesized with a smooth texture (**a**) and a rough model (**b**). The smooth texture is 0.1 mm high; its wavelengths in X and Y directions are 1.58 mm and 0.79 mm, respectively. The rough surface has RMS = 10 μm, Hurst exponent H = 0.5, and surface resolution dx = 20 μm. *q*_*r*_, *q*_*L*_ and *q*_*s*_ indicate roll-off, large and small wavelength cut-offs.

**Figure 2 polymers-13-01022-f002:**
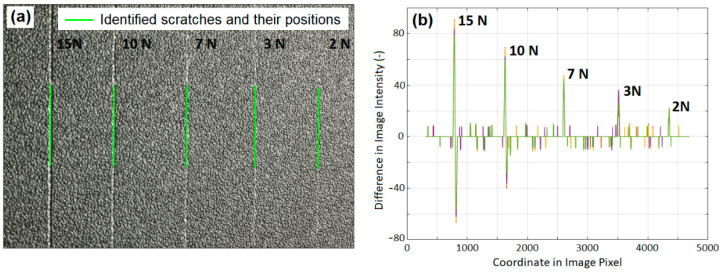
(**a**) Scratches identified and their positions (indicated by short green lines) on SB texture, and (**b**) Image intensity difference in contrast to background for calculating the properties of the scratches.

**Figure 3 polymers-13-01022-f003:**
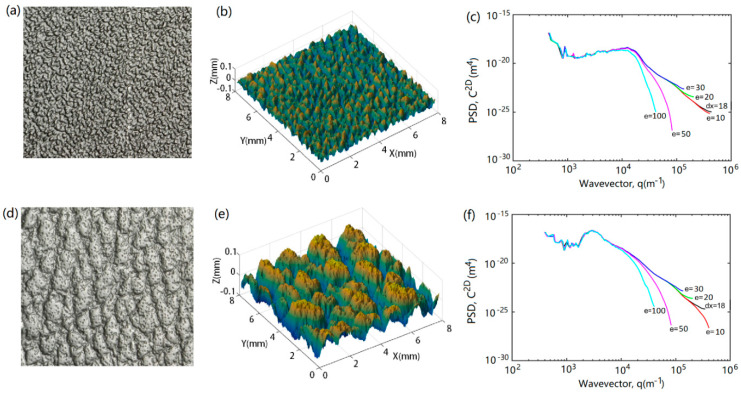
Images of textured surfaces of (**a**) shallow bump (SB), (**d**) coarse bump (CB), and (**b**,**e**) their respective local 3D scanned profiles and (**c**,**f**) radially averaged 2D PSD, C^2D^, for original sample (dx = 18 μm) and square-meshed surfaces (e = 10–100 μm). The surface is isotropic such that C^2D^ is radially symmetric, and the radial average is provided; dx is the average resolution size of the 3D scanned surfaces; e is the element size of surface meshes.

**Figure 4 polymers-13-01022-f004:**
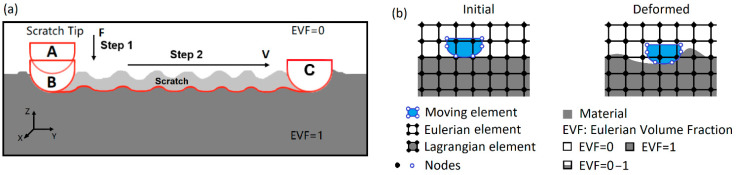
(**a**) Schematic of simulation model and (**b**) the concept of CEL method for scratching.

**Figure 5 polymers-13-01022-f005:**
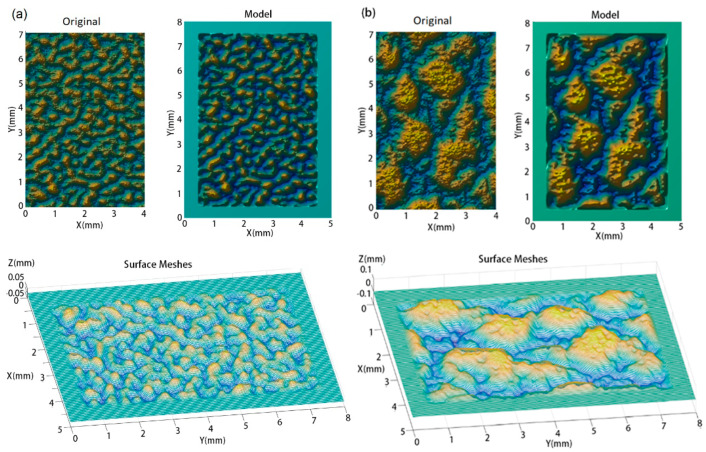
Original sample surface, model, and surface meshes for (**a**) SB texture, and (**b**) CB texture. The original surfaces include all small peaks. The models capture the texture features, small peaks and craters shown in the original surfaces. A 0.5 mm plain margin around the texture was created to extend the plaque model size.

**Figure 6 polymers-13-01022-f006:**
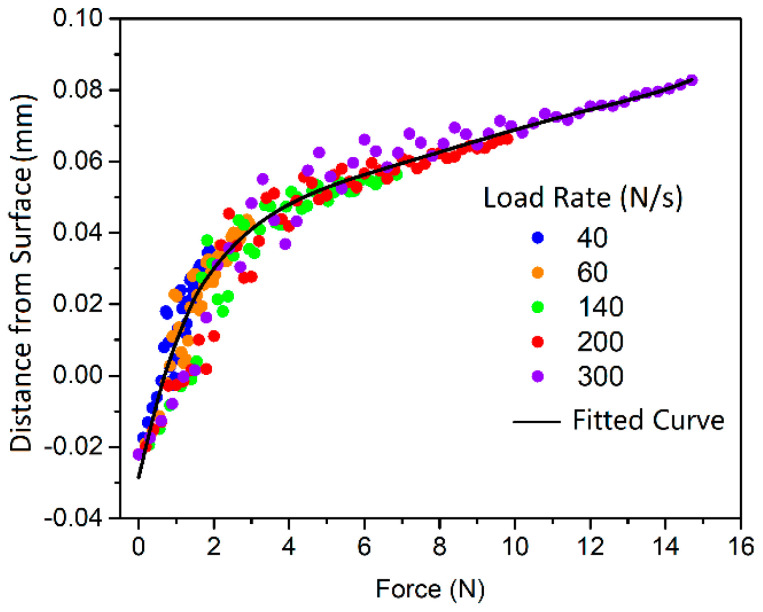
Force and indentation depth at different force-loading rates for rough surface.

**Figure 7 polymers-13-01022-f007:**
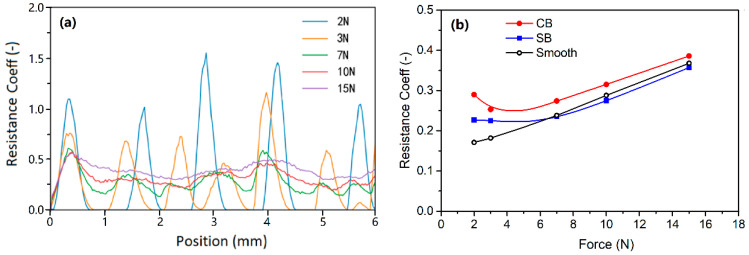
(**a**) Instantaneous resistance coefficient in scratching and (**b**) scratch resistance coefficients of different surfaces.

**Figure 8 polymers-13-01022-f008:**
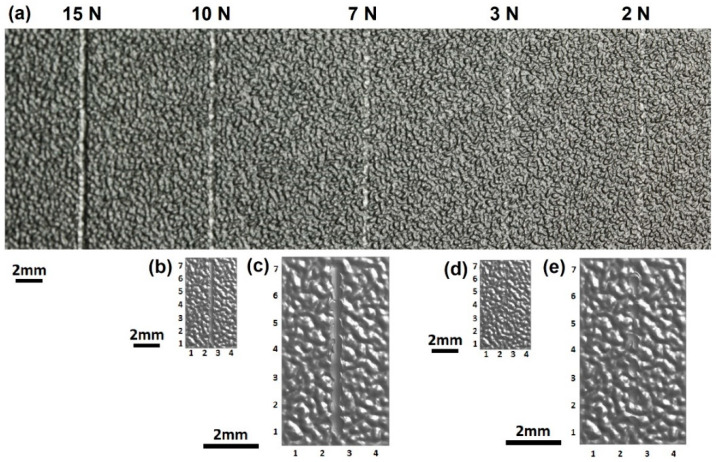
Images of scratches (**a**) produced by instrument under different forces and simulated under (**b**,**c**) 10 N and (**d**,**e**) 3 N on the SB texture. Images (**c**,**e**) are enlarged views of the simulated scratches presented in (**b**,**d**).

**Figure 9 polymers-13-01022-f009:**
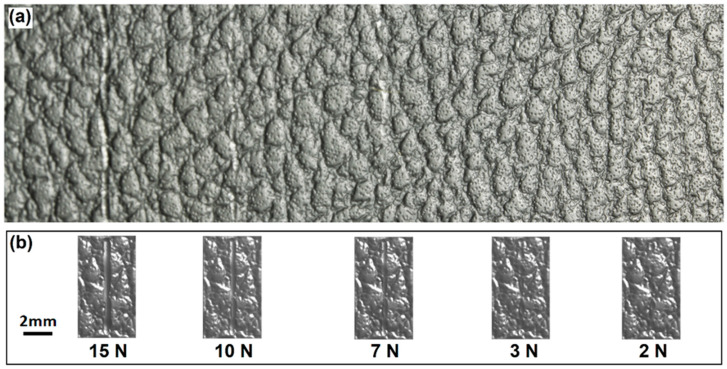
Images of scratches produced under different forces on the CB texture. (**a**) experiment and (**b**) simulation.

**Figure 10 polymers-13-01022-f010:**
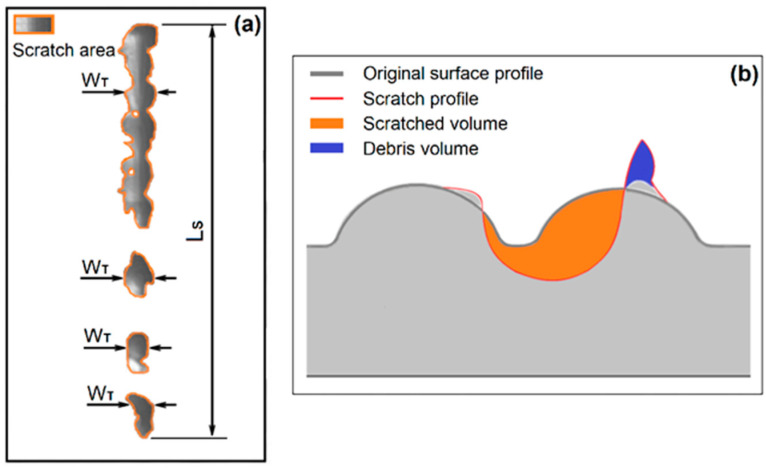
Evaluation parameters of scratches on a textured surface. (**a**) scratch size and (**b**) material damage. Figure b was adapted from [39].

**Figure 11 polymers-13-01022-f011:**
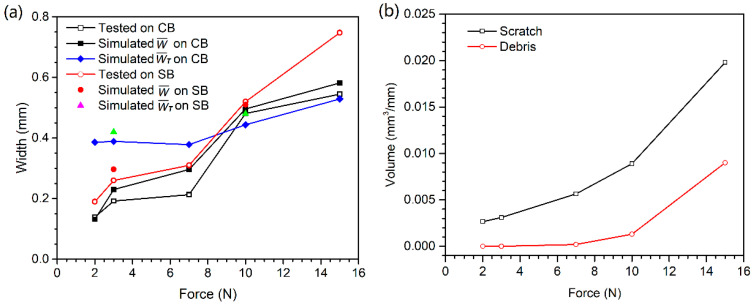
(**a**) Scratch size and (**b**) materials damage as a function of scratching force.

**Figure 12 polymers-13-01022-f012:**
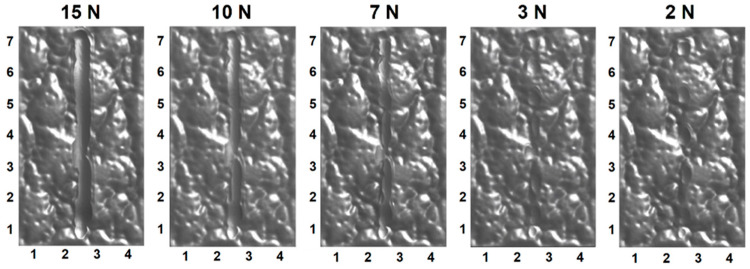
Simulated scratches showing the hidden bruise spots (2 N and 3 N) and continuous material damage (10 N and 15 N) of the CB textured surface.

**Table 1 polymers-13-01022-t001:** Parameters of surface features (in μm).

Texture	Shape of Grain	Max Height	Grain Size *	Grain Spacing *	R_A_
Shallow Bump	Sharp	0.070	213 (124)	460 (217)	3.56
Coarse Bump	Round	0.110	700 (374)	1856 (466)	2.52

* The data are the average values; standard deviations given in brackets.

**Table 2 polymers-13-01022-t002:** Yeoh and PRF constants for ADX-5017.

Yeoh Constants		PRF Constants					
C_10_	1.71396	S_1_	122.537	S_2_	62.0356	S_3_	12.579
C_20_	−0.20298	τ_Base1_	18.9996	τ_Base2_	15.8083	τ_Base3_	12.5455
C_30_	0.0174978	n_1_	14.2556	n_2_	14.5184	n_3_	11.6288
κ	2000	m_1_	−0.499632	m_2_	−3.01084 × 10^−4^	m_3_	−3.24107 × 10^−6^

## Data Availability

The data presented in this study are available on request from the corresponding author.

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
