# Peer review of "Finite Element Modelling and Experimental Validation of Scratches on Textured Polymer Surfaces"

_polymers, 2021, doi:10.3390/polym13071022_

Round 1

Reviewer 1 Report

Paper gives combination of experimental and calculation study of the scratcher on the texture polymers. Paper is well organized, results presentation is clear, and it can be accepted after some revision.

1) Please descibe, why you select namely ADX-5017 as test materials? Additionaly, you indicate it as simply polypoefin, whereas indeed it is polypropylen filled with talc and any impact modifies. PLease describe and discuss, how these fillers and additives can affect your experimental results.

2) PLease describe the experiental procedure, applied to create the surface textures on the polymer. Please also add in section 2 the table, summarizing all the created textures and their main parameters.

3) Fig. 2a. Differences between the scratches should be indicated in figure and described in the text. Fig. 2b. The position of this curve related to the surface presented in Fig. 2 a, should be given.

Author Response

Thank the reviewer for the comments.

  • The material ADX5017 was used in this study, as it is a widely used in automotive industries, including the collaboration partner of the project. We changed the first paragraph in experimental section to explain this implicitly. Regarding the suggestion on the discussion of the influence of the fillers and the additives, we believe this would make the manuscript more extensive and complete. As the manuscript focuses on the influence of the textured surface, the influence of the fillers and the additives was not considered in this work.
  • The first paragraph in experimental section was edited to explain the fabrication of the samples. In line 159, the scratch test method was introduced in the revised version. Together with the following context (lines 160-178), the experimental procedure should be clearly explained.

To explain the creation of the surfaces and the models, lines 261-263 were added to explain how the surface data were acquired. The generation of textured surfaces for models was explained.

Table 1 was added to provide the main parameters of the surfaces.

  • The figure (Fig2) was re-generated by including more information to address the comment.

Reviewer 2 Report

Abstract: 

The abstract reads well. However, it could contain some take-home messages and main conclusions, since it rather summarizes the conducted research

Graphical abstract: 

The graphical abstract is okay. However, it may contain too much information. In addition, it remains unclear what the x- and y-axis reflect. Please improve. 

Keywords: 

The keywords may be chosen in a better way.

Introduction: 

  • The introduction reads well. The language is fine. That is highly appreciated. It may contain some more recent review article about surface textures. There are some manuscript dealing with multi-scale surface textures from Greiner et al. and Grützmacher et al. for instance.
  • The presented literature review is particularly incomplete. There are studies that merge experimental and numerical results for textured surfaces (although considering different material systems). You may refer to some works from China. For your further guidance, you may want to check: 
  • --Nanosecond Pulsed Laser Ablation on Stainless Steel− Combining Finite Element Modeling and Experimental Work
  • --Nanosecond pulsed laser ablation of silicon—finite element simulation and experimental validation

Experimental: 

  • The first paragraph of the experimental section reads rather extensive. Please shorten.
  • Please explain why you have selected PSD to characterize the surface roughness. 
  • Please provide more information about the surface texturing

Results and discussion: 

  • Figure 3 and 4 should be combined.
  • In section 3.2, one gets the feeling that the experimental section is mixed with the section about results and discussion. Please check and organize. 
  • It would be nice if you could summarize the materials properties and constants used for the hyper elastic modeling. 
  • Please comment on the boundary conditions used for the scratch modeling. 
  • When looking at your Figure 6, one gets more the impression that you have been studying rough surfaces (stochastic surfaces) and not really surface textures, which are known to contain periodic, deterministic features. Please check and verify. It is fine to work on rough surfaces, but then the wording "textured surfaces" is misleading. 
  • How did you obtain the curve presented in Figure 7?
  • Please explain the periodic fluctuations present in Figure 8? Is that some numerical instability? or any surface roughness effect? Please verify and comment.
  • Figure 10 needs some scale bars. 
  • Are the results presented in Figure 9 statistically relevant or when considering an error bar, there would be no significant differences? 
  • Conclusions are too long and should be shortened as well as streamlined. 
  • Please present some error bars for Figure 14. 
  • It is recommended to combine Figure 13 and Figure 14. 
  • How do you explain the rather large deviations observable in Figure 13 with respect to the experimental and numerical results?
  • The discussion of the results should be extended. 
  • Please check the numbering of the references. The number appears twice. 

Author Response

Thank the reviewer for the comments
Response to the comments on Abstract: 
  • The abstract was re-edited by including some main finding in this study.
Response to the comments on Graphical abstract: 
  • The graphical abstract was improved based on the comment.
Response to the comments on Keywords: 
  • The keywords were changed.
Response to the comments on Introduction: 

Thank the reviewer for the comment.

  • Following the reviewer’s comments, the first paragraph was changed by including more literature relating to surface texturing to demonstrate that the technology has been extensively studied and widely applied. The revised paragraph also explicitly defined the scope of this study to limit the literature review and emphasize the content of the interest in this work.      
Response to the comments on ‘Experimental’: 
  • The comments were addressed by re-writing the first paragraph of the experimental section.
  • Lines 129-130 were added to explain the use of PSD.
  • Table 1 was added to provide the texture information.
Response to the comments on ‘Results and discussion: ‘
  • Figure 3 and 4 were combined, following the reviewer suggestion.
  • Thank the reviewer for the comment “In section 3.2, one gets the feeling that the experimental section is mixed with the section about results and discussion. Please check and organize.”

We checked the organization of the content of the manuscript, section 3.2 explained the generation of surface models and how to get desired features in the models. Section 4.1 in the results and discussion was organized to demonstrate the outcomes. The subtitles of sections 3.1 and 3.2 were then corrected to make the objectives of each section clear.

  • To address the reviewer comment on “It would be nice if you could summarize the materials properties and constants used for the hyper elastic modeling.” Table 2 was added to provide the details of the material property model.
  • Based on the comment on the boundary conditions used for the scratch modeling, Figure 4 was edited and the context (lines 314-316) was edited to explain the boundary conditions.

  • When looking at your Figure 6, one gets more the impression that you have been studying rough surfaces (stochastic surfaces) and not really surface textures, which are known to contain periodic, deterministic features. Please check and verify. It is fine to work on rough surfaces, but then the wording "textured surfaces" is misleading. 

Thank the reviewer for the comment. We agree that this could result in the misunderstanding of the definition of the surfaces. Some literatures such as “Structured, Textured or Engineered Surfaces” by the authors from National Institute of Standards and Technology, USA., give the definition of these terms. To clarify this issue, the manuscript was changed by including the manufacturing method of the texturing of polymer products (lines 35-36) and the method used in the sample fabrication in the study (in section 3.1).     

  • To clarify the question on how to obtain the curve presented in Figure 6 (Figure 7 in old version), the first paragraph in 4.2 was changed.

  • To explain the periodic fluctuations in Figure 7 (Figure 8 in old version), the second paragraph in 4.1 was edited, including the addition of lines 411-412, lines 413-415.

  • The scale bars in Figure 9 (Figure 10 in old version) were added. 

  • For the comments in the results presented in Figure 8 (Figure 9 in old version), the data were determined based on the forces from the simulation results of scratches. The data were examined. They are stable and do not change when the scratches are enough in length, so that the error would be no physical meaning in this case, although they are the average of the signals.   

  • The conclusions were rewritten thoroughly.

  • Regarding to the error bars in Figure 12 (b) (Figure 14 in old version), we checked the data. As the scratched volume and the debris volume refer to the total volumes of material deformation and damage, there is no error for both data.

  • Following the reviewer’s recommendation on combining Figure 13 and Figure 14, we tried to make one figure for all data, but it seems the figure created is too busy, so we finally joined the two pictures horizontally (as shown in Figure 12).

  • To explain the rather large deviations observable in Figure 12 (a) (Figure 13 in old version) with respect to the experimental and numerical results, we edited the manuscript by adding lines 545-547. This clarified the explanation of the deviation in this paragraph.   

  • Based on the suggestion, the discussion of the results was extended. The main changes include:
    • The second paragraph in 4.2, Lines 411-415
    • The second paragraph in 4.4, (Lines 500-506)
    • The third paragraph in 4.4, (lines 512 -518)
    • The last paragraph in 4.4

  • Thank the reviewer for finding the errors in the numbering of the references.

Round 2

Reviewer 1 Report

Paper can be accepted now

Author Response

Thanks the reviewer.

We read the manuscript thoroughly and made some language correction.

Reviewer 2 Report

Thank you very much for the nice revision and addressing most of my questions and concerns. The quality of the review has certainly improved. 

There are some additional, mostly minor aspects, which I would recommend to improve before the manuscript may be suitable for acceptance: 

  • The readability of Figure 1 should be improved. All individual figures are rather small. You may rearrange the position of the individual figures thus improving their readability.
  • Please extend on what a multi-finger scratch tester is...
  • It would be recommendable to present the parameters shown in Table 1 with mean values as well as error bars. 
  • You could also combine Figure 7 and 8. 
  • Figure 11 is too small. Please enlarge. 

Author Response

Thank the reviewer very much for the comments.

  • First, we read the manuscript thoroughly and made some language correction.

  • Based on the reviewer suggestion, Figure 1 was re-organized, and the PSD figure was reproduced by changing the style, colour and width of the lines to make the overlapped lines readability. The size of labels and axis-titles was also changed to improve the quality of the figure.
  • Details (Producer and model) of the multi-finger scratch tester used were provided in the revised version.
  • The standard deviations of the size and spacing of grains were added in Table 1 and explained under the table.
  • Figure 7 and 8 were combined and the corresponding context was edited.
  • Figure 10 (Figure 11 in old version) was enlarged.